# Genetic Testing to Predict Weight Loss and Diabetes Remission and Long-Term Sustainability after Bariatric Surgery: A Pilot Study

**DOI:** 10.3390/jcm8070964

**Published:** 2019-07-03

**Authors:** Andreea Ciudin, Enzamaria Fidilio, Angel Ortiz, Sara Pich, Eduardo Salas, Jordi Mesa, Cristina Hernández, Olga Simó-Servat, Albert Lecube, Rafael Simó

**Affiliations:** 1Institut de Recerca Vall d’Hebron, Universitat Autònoma de Barcelona (VHIR-UAB), 08035 Barcelona, Spain; 2CIBER de Diabetes y Enfermedades Metabólicas Asociadas, Instituto de Salud Carlos III, 08950 Barcelona, Spain; 3Scientific Department, Gendiag.exe, Joan XXIII, 10, Esplugues de LLobregat, 08950 Barcelona, Spain; 4Endocrinology and Nutrition Department, Hospital Universitari Arnau de Vilanova, IRBLleida, Universitat de Lleida, 25198 Lleida, Spain

**Keywords:** diabetes, obesity, bariatric surgery

## Abstract

Introduction: The aim of this pilot study was to assess genetic predisposition risk scores (GPS) in type 2 diabetic and non-diabetic patients in order to predict the better response to bariatric surgery (BS) in terms of either weight loss or diabetes remission. Research Design and Methods: A case-control study in which 96 females (47 with type 2 diabetes) underwent Roux-en-Y gastric by-pass were included. The DNA was extracted from saliva samples and SNPs were examined and grouped into 3 GPS. ROC curves were used to calculate sensitivity and specificity. Results: A highly sensitive and specific predictive model of response to BS was obtained by combining the GPS in non-diabetic subjects. This combination was different in diabetic subjects and highly predictive of diabetes remission. Additionally, the model was able to predict the weight regain and type 2 diabetes relapse after 5 years’ follow-up. Conclusions: Genetic testing is a simple, reliable and useful tool for implementing personalized medicine in type 2 diabetic patients requiring BS.

## 1. Introduction

Obesity represents a major public health problem and it is associated with a significant economic burden on the health systems of developed countries, mainly due to the associated co-morbidities. Among these co-morbidities, type 2 diabetes (T2D) is one of the most important.

Bariatric surgery (BS) is a successful treatment for morbid obesity and leads to a dramatic improvement in obesity-related comorbidities [1]. The remission rate of T2D after BS is around 60–70% after 1 year of follow-up [2]. Therefore, there is a significant proportion of non-responders to BS in terms of diabetes remission. Additionally, after 5 years, there is about a 20–35% relapse of T2D after Y-de-Roux gastric by-pass (RYGB) [3,4,5]. A score based on clinical variables for the pre-operative prediction of T2D remission following RYGB surgery (DiaRem) was proposed [6]. However, this model has several limiting factors [7] and it has not been generally adopted in clinical practice. At present, there are no reliable predictors of T2D remission and relapse after BS.

In recent years, interest in the genetic influence on the response of different treatments for obesity has increased. Two retrospective studies [8,9] showed that several single nucleotide polymorphisms (SNPs) were associated with a poor response to BS. However, in these studies the discrimination capacity of the GPS was not significant, and the role of T2D in the response to BS and the impact of these genetic factors on diabetes remission were not evaluated.

On this basis, the aim of the present study was to evaluate whether genetic markers can be used for the prediction of adequate weight loss and diabetes remission after BS.

## 2. Material and Methods

A single-center, retrospective observational pilot study in a third-level university hospital (Vall d´Hebron University Hospital, Barcelona, Spain) was conducted following the Strengthening the Reporting of Observational Studies in Epidemiology guidelines. The study comprised patients that underwent RYBG surgery between January 2010 and December 2012. The inclusion criteria were women, stable weight in the prior 6 months before BS, and minimum of 5 years of follow-up after BS. In order to avoid heterogeneity and given that the vast majority of the patients under bariatric surgery were women, we decided to rule out the inclusion of men in this pilot study. The patients were informed about the study and they all signed the written informed consent form.

The exclusion criteria were male, marked mobility problems, a different BS technique apart from RYBG, and severe psychiatric or eating disorders. For the genetic study, a sample of saliva was collected. The characteristics of RYBG were food loop length: 150–180 cm, and bilio-pancreatic loop length: 120 cm, gastric pouch 30 cc^3^. The technique was the same in all cases, performed by the same surgical team in our hospital.

Excess body weight (EBW) was defined as the amount of weight that was in excess of the ideal body weight (IBW). The percentage of excess weight loss (EWL) was calculated according to the formula: %EWL = (weight before BS (kg) − weight after BS (kg)/EBW(kg)) × 100. The post-BMI weight regain was defined as a 10% regain of the minimal weight after BS. The minimal weight after BMI was achieved at 2 years follow-up for all of the patients.

Diabetes remission was defined according to American Diabetes Association (ADA) criteria [10]. Relapse of T2D was defined as one or more of the following conditions: (a) restarting diabetes medication; (b) one or more HbA1c measures ≥ 6.5%; and/or (c) one or more fasting glucose measures ≥ 126 mg/dL [11].

The study was approved by the Local Ethics Committee and registered at Clinical.Trials.gov, NCT02405949.

### 2.1. Genotyping and Sequencing

The DNA was extracted from saliva samples and processed by GoldenGate^®^ Genotyping Assay for VeraCode. The genetic predisposition was assessed using Nutri inCode (NiC) (Ferrer inCode) and selecting the 57 SNPs associated with susceptibility to diabetes, obesity, appetite regulation, weight loss in response to hypocaloric diet, and the response to BS. The details about the SNPs are reflected in the Appendix A. The selected SNPs were grouped into three genetic predisposition risk scores (GPS): diabetes remission, weight loss in non-diabetic subjects, and weight loss in subjects with diabetes.

### 2.2. Statistical Analysis

In order to assess the best predictive GPS, patients were distributed into 4 subgroups according to the BS response (%EWL) and the presence of T2D: (1) %EWL < 40% without diabetes (*n* = 15); (2) %EWL < 40% with diabetes (*n* = 16); (3) %EWL > 75% without diabetes (*n* = 35); and (4) T2D and %EWL > 75% with diabetes (*n* = 31). Univariate and multivariate logistic regressions were used to establish associations. Akaike Information Criterion (AIC)-based backward selection was used to remove insignificant terms from an initial model containing all the candidate predictors. The calibration of the model’s adequacy was determined by the Hosmer–Lemeshow test. The area under the ROC curve (AUROC) was used for evaluating the prediction performance of the models. The cut-offs for the developed algorithms were selected as the point which maximizes the Youden index.

## 3. Results

The clinical characteristics of the patients included in the study are shown in Table 1. Apart from age, we did not find any significant differences between diabetic and non-diabetic subjects before BS. The diabetic treatment received by subjects with diabetes is displayed in Figure 1. No other medication apart from AINEs occasionally and vitamin supplements as per protocol after bariatric surgery (ciancobalamin 1000 mcg/month, colecalciferol 25.000–100.000 UI/month) were administered.

In the subgroup of the non-diabetic patients, the multivariate logistic regression equation for predicting positive weight loss response (%EWL > 75%) after the BS (NiC-Bariatric-ND) includes SNPs associated with weight loss in response to a hypocaloric diet and SNPs associated to appetite regulation. The model showed an AUROC of 0.763 (95% CI 0.605 to 0.920; *p* < 0.001), a sensitivity of 86.49%, and a specificity of 57.14%. The calibration of the adequacy of the model determined by the Hosmer–Lemeshow test was 0.679.

The continuous variables were median (1st quartile; 3rd quartile) and the categorical data were percentages. BMI: body mass index. EWL: excess of weight loss. BS: bariatric surgery. Hypertension was defined by increased systolic (≥140 mmHg) or increased diastolic (≥90 mmHg) blood pressure or by the use of antihypertensive drugs, according to current guidelines. Dyslipidemia was defined by the use of lipid-lowering drugs, decreased values of HDL cholesterol (men < 0.9 mmol/L, women < 1.0 mmol/L) or by at least one increased value of total cholesterol (>5.2 mmol/L), LDL cholesterol or triglycerides (>1.7 mmol/L).

Weight regain after 5 years’ follow-up was seen in 9.6% of the patients. The model to identify the patients who had presented weight regain after 5 years’ follow-up showed an AUROC of 0.834 (95% CI 0.705 to 0.923; *p* < 0.0001), a sensitivity of 100%, and a specificity of 70.21%. The calibration of the adequacy of the model determined by the Hosmer–Lemeshow test was 0.5148.

In T2D patients, the multivariate logistic regression equation for the prediction of weight loss response (%EWL > 75%) after BS (NiC-Bariatric-D) included SNPs associated with weight loss in response to hypocaloric diet, SNPs associated to response to BS [9], and SNPs associated to response to lifestyle interventions [11]. The model showed an AUROC of 0.929 (95% CI 0.850 to 0.99; *p* < 0.001), a sensitivity of 87.10% and a specificity of 93.33%. The calibration of the model’s adequacy determined by the Hosmer–Lemeshow test was 0.291. Weight regain in subjects with diabetes was observed in 17.5% of them. The model to identify patients with diabetes who will regain weight after a follow-up of 5 years after bariatric surgery showed in this case an AUROC of 0.781 (95% CI 0.623 to 0.896; *p* < 0.04), a sensitivity of 71.43%, and a specificity of 84.85%. The calibration of the model’s adequacy determined by the Hosmer–Lemeshow test was 0.8664. Figure 2 shows the AUROC corresponding to weight regain in the whole (Figure 2A) population, non-diabetic subjects (Figure 2B), and T2D patients (Figure 2C).

Diabetes remission was seen in 73.91% of the type 2 diabetic patients included in the study (66.67% in the group of %EWL < 40% and 77.42% in the group of %EWL > 75%). Diabetes relapse was seen in 25% of the patients.

The multivariate logistic regression equation for the prediction of diabetes remission and relapse after BS (NiC-Bariatric-DR) included SNPs associated with obesity, SNPs associated with weight loss in response to hypocaloric diet, SNPs associated with appetite regulation, and SNPs associated with genetic predisposition to diabetes. This prediction model showed an AUROC of 0.868 (95% CI 0.709 to 0.976; *p* < 0.0001) for diabetes remission, with a sensitivity of 76.47% and a specificity of 83.33%. In our population, the AEROC for DiaRem was lower than obtained by genetic testing, (0.69 versus 0.86), and when both scores were combined, the AUROC was 0.87, with a sensitivity of 88.49% and a specificity of 80% (Figure 3).

Regarding diabetes relapse after 5 years, the model based showed an AUROC of 0.833 (95% CI 0.682 to 0.932; *p* < 0.0001), with a sensitivity of 90.00 and a specificity of 80.00 (Figure 4). The calibration of the adequacy of the model determined by the Hosmer–Lemeshow test was 0.280.

## 4. Discussion

Bariatric surgery provides adequate and sustainable weight loss and T2D remission, but 15–20% of the subjects do not reach these targets [12]. A recent study [13] showed a high inter-individual variability of the EWL response at mid-term after BS and that poor EWL could be illustrated by two different patterns: poor sustained weight loss or pronounced weight regain. At present, there are no reliable biomarkers for individual response to BS. Due to the increasing availability of BS around the world and the alarming prevalence of obesity and its associated co-morbidities such as T2D, the discovery of biomarkers that will permit us to identify the best candidates for BS are urgently needed.

In the present study, we developed genetic-based algorithms for the prediction of %EWL after BS and for T2D remission with high sensitivity and specificity. Still et al. [8] proposed a genetic score to predict the %EWL after BS, showing a non-statistically significant AUROC, a sensitivity of 48.39, and a specificity of 73.33. In addition, this score did not take into the account the presence of diabetes.

Regarding diabetes remission after BS, we analyzed for comparison purposes the predictive capacity of DiARem scores [8] in our study population. In our population, the AUROC for DiaRem was lower than obtained by genetic test (0.69 versus 0.86), and when both scores were combined, the AUCROC was 0.87, with a sensitivity of 88.49% and a specificity of 80.00%. This finding supports the use of genetic testing in clinical practice.

It is worth mentioning that in diabetic patients, the rate of remission was not significantly different between the group with %EWL < 40% and the group with %EWL > 75% (*p* = 0.674). In addition, previous data showed that about 30% of T2D patients that are able to discontinue the medication after BS will present a relapse within the first 5 years [3,4,5]. Some studies found weak correlation between weight regain, younger age or lower BMI before BS as predictors of T2D relapse after BS [5,14], while other studies found no association [3]. Therefore, at present, there are no reliable predictors of T2D relapse after BS. In our study, the proposed score showed a high predictive value of T2D relapse after BS, thus, underlying the potential key role of genetic testing in precision medicine in order to assure better outcomes after BS. Interestingly, in our study, in the subgroup of T2D patients, the inclusion of SNPs associated to response to BS did not improve the prediction scores, suggesting that these genes are not critical or do not intervene in the remission and relapse of T2D. This finding suggests that the physiopathology of diabetes remission and relapse after RYGB might not be related with the %EWL in this population.

Overall, these results are intriguing and point to a genuine genetic background in the mechanisms involved in diabetes remission and relapse after BS, perhaps related to insulin resistance.

In conclusion, in this pilot study we have developed highly sensitive and specific genetic predictive scores of responses to BS in terms of weight loss and T2D remission and the long-term sustainability of these effects. These results would allow us not only to implement a more effective and personalized BS, but also to optimize healthcare resources. However, further studies with a larger sample size to confirm this pilot study are needed.

## Figures and Tables

**Figure 1 jcm-08-00964-f001:**
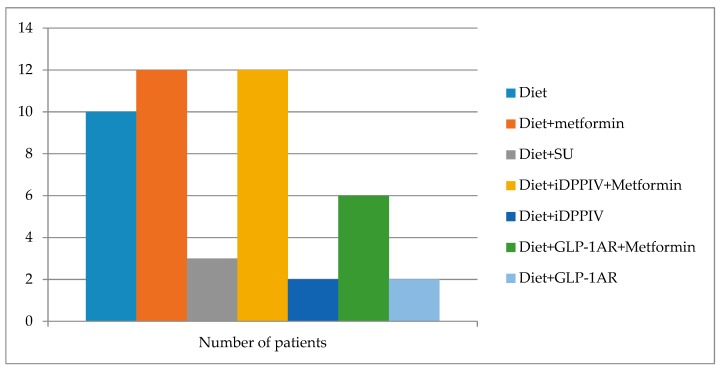
The diabetic treatment received before BS by the subjects with diabetes included in the study. SU: sulphonylurea, iDPPIV: DPPIV enzyme inhibitor, GLP-1AR: GLP-1 receptor agonists.

**Figure 2 jcm-08-00964-f002:**
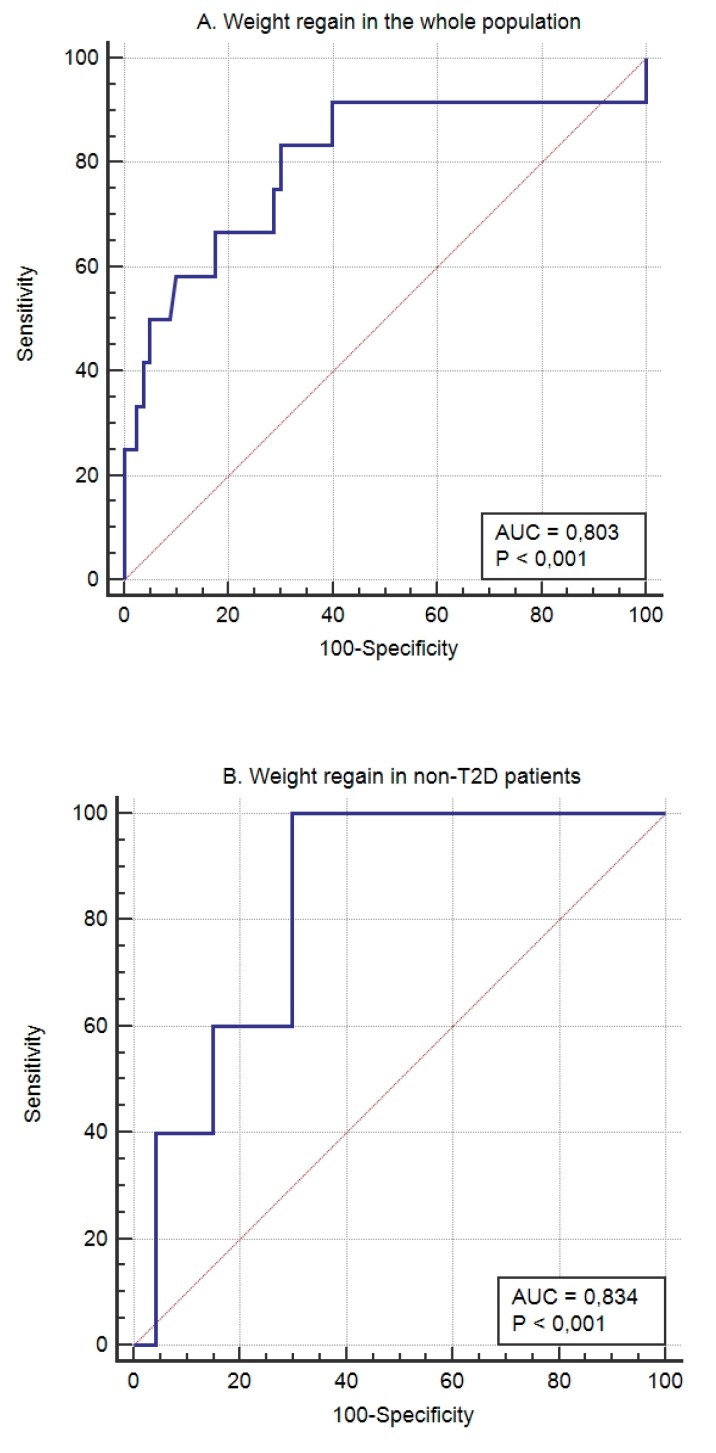
The predictive capacity of the genetic score for weight regain after 5 years’ follow-up in the whole (**A**) population, non-T2D subjects (**B**), and T2D patients (**C**).

**Figure 3 jcm-08-00964-f003:**
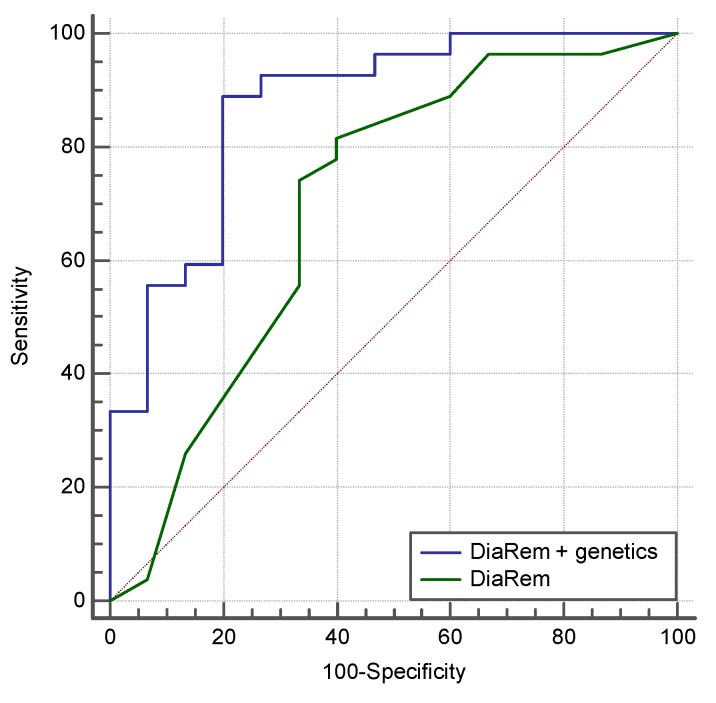
The predictive capacity of the DiARem score and the combination between DiARem and genetics in our study population. The AUROC for DiaRem was lower than obtained by genetic test (0.69 versus 0.86), and when both scores were combined the AUCROC was 0.87, with a sensitivity of 88.49% and a specificity of 80.00%.

**Figure 4 jcm-08-00964-f004:**
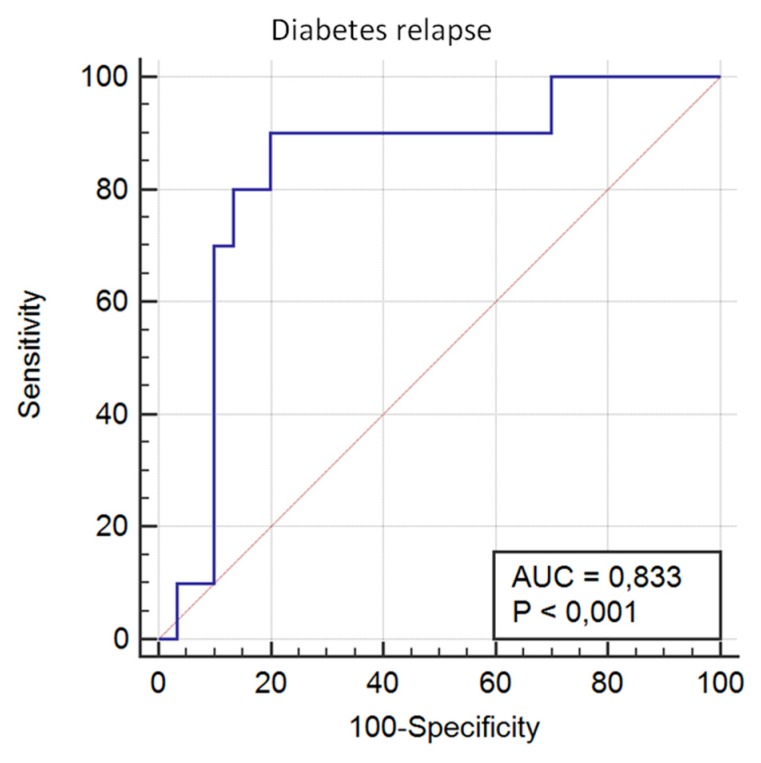
The predictive capacity of the genetic score for T2D relapse after 5 years’ follow-up.

**Table 1 jcm-08-00964-t001:** Baseline characteristics of the patients included in the study.

	Non-Diabetic Patients	Type 2 Diabetic Patients	*p*
*N*	50	47	
Age (years)	48.0 (37.5; 55.0)	52.0 (46.0; 58.8)	0.0016
Initial BMI (Kg/m^2^)	45.2 (43.0; 48.5)	42.5 (40.1; 46.4)	0.008
2 y post-BS BMI (Kg/m^2^)	31.8 (26.1; 35.6)	30.9 (26.8; 35.7)	n.s.
5 y post-BS BMI (Kg/m^2^)	32.63 (21; 52.14)	33.68 (21; 46.43)	n.s.
Hypertension (%)	48.3	49.5	n.s.
Dyslipidemia (%)	43.2	45.7	n.s.
Sleep apnea (%)	27.2	29.7	n.s.

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
