# Peer review of "Genetic Testing to Predict Weight Loss and Diabetes Remission and Long-Term Sustainability after Bariatric Surgery: A Pilot Study"

_jcm, 2019, doi:10.3390/jcm8070964_

Round 1
Reviewer 1 Report
thank you for your revisions
Reviewer 2 Report
I have looked at the new manuscript now and the authors have included the information I think is necessary in the supplementary appendix.In my opinion, the article might be accepted as it is now
This manuscript is a resubmission of an earlier submission. The following is a list of the peer review reports and author responses from that submission.
Round 1
Reviewer 1 Report
The study examines an interesting topic; whether genetic markers can be used for the prediction of adequate weight loss and diabetes remission after BS.
My main concern is that too few details are given about the SNPs used: Why have you chosen these 57, which SNPs are used in which of the three GPSs, and which SNPs are included in each multivariate logistic regression equation.
It is easy to create a model with a high AUROC for any phenotypic trait when you start with many SNPs, but most of these GPSs will not be reproduceable in other cohorts. It is therefore fundamental to provide enough details that other researchers can try to reproduce your findings.
I also miss information about quality control - could the results of the genetic test be used in all the 97 subjects or were some included based on the quality of the genotyping? Did you use any imputation technique?
Author Response
The study examines an interesting topic; whether genetic markers can be used for the prediction of adequate weight loss and diabetes remission after BS.
My main concern is that too few details are given about the SNPs used: Why have you chosen these 57, which SNPs are used in which of the three GPSs, and which SNPs are included in each multivariate logistic regression equation.
Answer: We used the genetic variants included in a nutrigenomic commercial product; Nutri inCode. The variants included in Nutri inCode were selected from published GWAS studies or replicated studies related to the genetic susceptibility to develop any of the phenotypes of interest for which GPS were constructed (diabetes, obesity, appetite regulation, weight loss in response to hypocaloric diet and the response to bariatric surgery). Nutri inCode was developed in collaboration with Professor Jose Maria Ordovas (Tufts University) a world-wide reputed scientist in the field of nutrition, nutrigenomics and atherosclerosis.
The SNPs used and the algorithms used for the construction of the GPS is a trade secret. Nevertheless, we are open to sharing that information with any researcher under a Material Transfer Agreement. This information has been included in the publication.
It is easy to create a model with a high AUROC for any phenotypic trait when you start with many SNPs, but most of these GPSs will not be reproduceable in other cohorts. It is therefore fundamental to provide enough details that other researchers can try to reproduce your findings.
Answer: As the SNPs are coming from GWAS and/or replicated studies we understand that the results could be reproducible in other cohorts. The company that developed Nutri inCode has developed other predictive clinical-genetic tests. They are about to publish a paper stating that two other predictive models constructed in a similar manner are statistically reproducible in other cohorts, and furthermore that predictive capability can even be improved if the algorithm is fine-tuned to the population of interest.
I also miss information about quality control - could the results of the genetic test be used in all the 97 subjects or were some included based on the quality of the genotyping? Did you use any imputation technique?
Answer: We could genotype all the variants of interest in the 97 subjects studied. Therefore, no imputation technique was used.
The authors have to be applauded for picking up a convincing idea with the possibility to change the field of bariatric surgery, especially in light of an emerging operative treatment of t2dm.
This pilot study involving 96 morbidly obese, female patients shows genetic testing to be reliable in prediction of t2dm remission and weight loss. It is clear that those results will have to stand the test of a prospective trial. (is there anything underway?)
I have a few remarks and questions:
Why those exclusion criteria? Why only females?
RYGB: what kind of RYGB was performed? Even though the procedure per se is quite standardized, the limb lengths vary enormously and this could have an influence on your data
The study duration lasted from 2010 til 2012: why wait so long to publish the results? Even further, this allows you for reporting another very interesting question you brought up in your manuscript yourself: weigh regain and relapse of t2dm (usually after 5y)
Two retrospective studies didn’t show a significant discrimination capacity, what is different in your study?
Definition of t2dm: there is a difference between insulin-dependent and non-insulin-dependent t2dm, duration of therapy also plays a role. Please state those data in table 1. It might explain the observation you mentioned on line 151ff
Did the patients take in any other medication, steroids,…?
Author Response
Dear Reviewer, first of all we would like to thank you for the constructive criticism, because we feel that the changes performed as consequence of the points raised by you have significantly improved the quality of the manuscript and we trust you find this manuscript suitable for publication. We are sorry for the delay, but an extention period was concede by the editor.
The authors have to be applauded for picking up a convincing idea with the possibility to change the field of bariatric surgery, especially in light of an emerging operative treatment of t2dm.
This pilot study involving 96 morbidly obese, female patients shows genetic testing to be reliable in prediction of t2dm remission and weight loss. It is clear that those results will have to stand the test of a prospective trial. (is there anything underway?)
Answer: Yes, we are including more patients and we are also planning a prospective study
I have a few remarks and questions:
Why those exclusion criteria? Why only females?
Answer:We have included only women since at that time 95% of the patients under bariatric surgery were women
RYGB: what kind of RYGB was performed? Even though the procedure per se is quite standardized, the limb lengths vary enormously and this could have an influence on your data.
Answer: The kind of RYGB was: food loop length: 200 cm, bilio-pancreatic loop length: 100 cm, common loop lengh: 150 cm. The technique was the same in all cases, performed by the same surgeon team, in our hospital.
The study duration lasted from 2010 til 2012: why wait so long to publish the results? Even further, this allows you for reporting another very interesting question you brought up in your manuscript yourself: weigh regain and relapse of t2dm (usually after 5y)
Answer: We recruited patients who underwent bariatric surgery 2010 to 2012 and were followed up in our Endocrine Department. The study was started in 2016 and although at that time we already had the clinical data of the two-year follow-up we had to wait for each patients’ annual visit to collect the saliva sample for the genetic study.
We have now analyzed the relapse of diabetes and this is included in the paper. The algorithm can also predict the relapse of diabetes and this capacity is improved if the initial and the two-years BMI is included in the algorithm.
Weight regain after 5 years was analyzed and included in the paper. At present there is no clear criteria defining the significant weight regain after bariatric surgery. There are different definitions and data in literature in heterogeneous. In our study we have considered weight regain if 10% of the minimum weight was regained at 5 years follow-up because this was the minimum weight regain associated with comorbidities relapse in our serie.
Two retrospective studies didn’t show a significant discrimination capacity, what is different in your study?
Answer: The main difference is the genetic variants included.
Definition of t2dm: there is a difference between insulin-dependent and non-insulin-dependent t2dm, duration of therapy also plays a role. Please state those data in table 1. It might explain the observation you mentioned on line 151ff
Answer: We do not have any insulin-dependent diabetic patient. We do have non-insulin-dependent diabetic patients treated with insulin. The use of insulin is considered in the DiaRem score.
Did the patients take in any other medication, steroids,…?
Answer:No, the only additional drug was AINEs and vitamins supplements as per protocol after bariatric surgery (ciancobalamin 1000mcg/month, colecalciferol 25.000-100.000 UI/month).
Round 2
Reviewer 1 Report
Thank you for your kind reply.
In my opinion, this paper should not be accepted without presentation of the specific SNPs in the paper itself; transparency about such findings is important to advance science.
Reviewer 2 Report
Thanks a lot for your revisions. I have two issues:
the minor one is the way RYGB is described: it is simply not possible that every subject in this paper has the same limb lengths, it's either a proximal or a distal RYGB with respective lengths
i think the revisions should be incorporated more in the text and not only in answers to reviewers